# Population Pharmacokinetics Modelling and Simulation of Mitotane in Patients with Adrenocortical Carcinoma: An Individualized Dose Regimen to Target All Patients at Three Months?

**DOI:** 10.3390/pharmaceutics11110566

**Published:** 2019-10-31

**Authors:** Yoann Cazaubon, Yohann Talineau, Catherine Feliu, Céline Konecki, Jennifer Russello, Olivier Mathieu, Zoubir Djerada

**Affiliations:** 1Department of Medical Pharmacology, EA3801, SFR Cap-Santé, Reims University Hospitals, 51100 Reims, France; ycazaubon@chu-reims.fr (Y.C.); yohann.talineau@gmail.com (Y.T.); cfeliu@chu-reims.fr (C.F.); ckonecki@chu-reims.fr (C.K.); 2Department of Medical Pharmacology, Montpellier University Hospitals, 34000 Montpellier, France; j-russello@chu-montpellier.fr (J.R.); o-mathieu@chu-montpellier.fr (O.M.)

**Keywords:** mitotane, adrenocortical carcinoma, pharmacokinetics, simulation, modelling, optimization

## Abstract

Mitotane is the most effective agent in post-operative treatment of adrenocortical carcinoma. In adults, the starting dose is 2–3 g/day and should be slightly increased to reach the therapeutic index of 14–20 mg/L. This study developed a population PK model for mitotane and to simulate recommended/high dosing regimens. We retrospectively analyzed the data files of 38 patients with 503 plasma concentrations for the pharmacokinetic analysis. Monolix version 2019R1 was used for non-linear mixed-effects modelling. Monte Carlo simulations were performed to evaluate the probability of target attainment (PTA ≥ 14 mg/L) at one month and at three months. Mitotane concentration data were best described by a linear one-compartment model. The estimated PK parameters (between-subject variability) were: 8900 L (90.4%) for central volume of distribution (V) and 70 L·h^−1^ (29.3%) for clearance (Cl). HDL, Triglyceride (Tg) and a latent covariate were found to influence Cl. The PTA at three months for 3, 6, 9, and 12 g per day was 10%, 55%, 76%, and 85%, respectively. For a loading dose of 15 g/day for one month then 5 g/day, the PTA in the first and third months was 57 and 69%, respectively. This is the first PKpop model of mitotane highlighting the effect of HDL and Tg covariates on the clearance as well as a subpopulation of ultrafast metabolizer. The simulations suggest that recommended dose regimens are not enough to target the therapeutic threshold in the third month.

## 1. Introduction

Mitotane (o,p’DDD) is an adrenolytic drug mainly used for adrenocortical carcinoma (ACC) for decades [1]. Currently, no available pharmacological options are better than mitotane. Although it has been used for long time, many of its pharmacological properties, such as exact pharmacodynamics and activation in humans, remain to be more explored and elucidated for better optimization [2]. Recently, Sbiera et al. found that mitotane inhibited sterol-*O*-acyl transferase 1 leading to impaired steroidogenesis and lipid induced endoplasmic reticulum stress [3]. Moreover, cholesterol is a key of steroid biosynthesis. Eisenhauer et al. showed that the combination of mitotane and statins is associated with a better tumour control according to the Response Evaluation Criteria In Solid Tumours criteria (RECIST) [4]. Boulate et al. investigated the effects of association between mitotane and statins in NCI-H295R human ACC cells and found that rosuvastatin potentiated the effects of mitotane which may provide novel therapeutic strategies for ACC [5]. This in vitro study was done with mitotane concentrations of 50 µM corresponding to the therapeutic index of plasma levels of 14–20 mg/L [6]. Furthermore, the ENS@T group analyzed the link between tumour response and plasma mitotane. The results of this study suggested maintaining plasma mitotane concentrations between 10 and 20 mg/L, and for patients without objective tumour response having a good tolerability, a target of 20–30 mg/L could be proposed [7]. Unfortunately, mitotane is hampered by the following unfavourable pharmacokinetic properties [8]: low oral bioavailability of 30–40% [9], very high volume of distribution due to its high lipophilicity (logP = 6.11 [10]) explaining its capacity to concentrate in deep adipose tissue compartment [11,12], and its elimination half-life varying between 18–159 days [13]. Consequently, reaching the optimal plasma concentrations takes months (1–6) after initiation. The best strategy to reduce this delay of target attainment remains debated [14]. The individual dose adjustment is still empirical, while the pharmacokinetic variability of mitotane appears to be very important [15]. Under these conditions, dosage adjustment must be rationalized using bayesian methods. Currently, there are two population pharmacokinetics studies describing mitotane plasma levels [15,16] but only one using a one-stage approach [17] to characterize interindividual variability [16]. To identify metabolism-related variability, it has to be noted that mitotane is progressively metabolized in the liver by hydroxylation and oxidation to two compounds: o,p’-DDE and o,p’-DDA [18,19]. Kitamura et al. identified Cytochrome P450 (CYP) isoforms which are implicated in the mechanism of dechlorination by microsomes from specific human cells expressing CYPs 1A1, 1A2, 2A6, 2B6, 2C9, 2D6, 2E1, and 3A4. It appeared that 3A4 and 2B6 were the most involved [19].

As mitotane is a strong inducer of CYP3A4 [20], potentially via the pregnane X receptor [21], Arshad et al. hypothesized that mitotane metabolism may be affected by autoinduction and modelled this by a linear enzyme autoinduction process. After CYP3A4, which shows limited genetic variability in the Caucasian population [22], the most important cytochrome involved in mitotane metabolism is CYP2B6 [19,22]. D’Avolio et al. found that a SNP of the CYP2B6 enzyme was involved in the variation of mitotane concentrations after three months of treatment. Patients carrying the wild-type ‘G’ allele showed significantly lower mitotane concentrations than those having the ‘T’ allele. However, after nine months, treatment difference was no more statistically significative. Previously mentioned for CYP3A4 and for drugs mediated by PXR, mitotane may progressively induce CYP2B6 [23,24].

The aims of this study were (i) to develop a population PK data analysis for mitotane administered by oral route in patients with adrenocortical carcinoma and (ii) to evaluate the probability of target attainment (≥14 mg/L) of 3, 6, 9, and 12 g/day at three months and propose a loading dose strategy targeting optimal plasma concentrations at 1 month.

## 2. Materials and Methods

### 2.1. Study Design and Subjects

This retrospective study was a population PK analysis of pooled data collected between 2008 and 2016 from patients treated with mitotane by oral route and total daily dose of 1–7.25 g/day in 2, 3, or 4 administrations. The two datasets were collected in patients treated for adrenocortical carcinoma in the university hospital of Reims (*n* = 25) and Montpellier (*n* = 13). Mitotane plasma concentrations were measured under recommended therapeutic drug monitoring conditions (TDM) as specified in summary of product characteristics. The inclusion criteria were patients with ACC, with therapeutic initiation, with at least four blood samples, and without missing data on covariates. The clinical ethical review committee was consulted and pronounced itself in favor of the analysis and the CNIL (“Commission Nationale de l’Informatique et des Libertés”) provided us with the registration number 2049775 v 0 (27 march 2018).

### 2.2. Assay Method

Mitotane was identified and quantified in plasma by liquid chromatography (ultraviolet method) [25].

### 2.3. PK Analysis

Population PK analysis of mitotane in plasma was conducted using nonlinear mixed effects modelling approach within the Monolix software (version 2019R1; Lixoft, Antony, France, http://lixoft.com/) implementing the stochastic approximation expectation maximization (SAEM) algorithm [26,27]. Diagnostic plots were performed by R software (version 3.3.0; R Development Core Team, Vienna, Austria, http://www.r-project.org/) using R package ggplot.

### 2.4. Basic Model Building

One-, two-, and three-compartment models with first-order absorption and elimination were initially compared. One- and two-compartment models with different equations to model a time-varying Cl were tested (see Appendix A files Appendix A ).

All individual parameters were considered to be log-normally distributed. The bioavailability (F) seemed to be variable, and as no intravenous data were available, we did not estimate it but fixed it at 35% in regards to the literature [9]. We did the same for first-order absorption constant (Ka): 24 day^−1^. Exponential random effects were consisted to describe between-subject variability. To describe the residual variability (ε), we assessed several error models (constant, proportional, or combined error model). The most appropriate model was selected based on the following criteria: Bayesian information criterion (BIC), usual diagnostic plots (GOF plots), and relative standard errors (RSE).

### 2.5. Covariate Analysis

From the basic model (without covariate), the effect of the following nine covariates on mitotane PK parameters was evaluated: age, gender, total body weight (TBW), ideal body weight (IBW) [28], lean body weight LBW) [28], body mass index (BMI) [28], high density lipoprotein (HDL), low density lipoprotein (LDL), and triglyceride (TG) levels. As variation of covariates were not significate, we took the median of each for each individual.

For continuous covariates, the parameter-covariate relationships were included as follows:(1)$$Cl_{i}=Cl_{pop}\times {\left(\frac{COV_{i}}{COV_{median}}\right)}^{\beta }\times e^{\eta_{Cl,i}},$$where *β* is the covariate effect to be estimated, *COV_i_* is the covariate value for subject *i*, and *COV_median_* is the median value of the covariate in the study population.

For binary covariates, the general equation was:(2)$$Cl_{i}=Cl_{pop}\times e^{\beta .COVi}\times e^{\eta_{Cl,i}},$$where *COV_i_* is 0 or 1 (0: Male; 1: Female). 

The unit of β coefficients is the logarithm of the unit of the associated parameter.

Moreover, D’Avolio et al. [24] showed differences on plasma concentrations between GG and GT/TT patients concerning the 516 polymorphism in the CYP2B6 gene (*rs3745274* (*G516T*)) Regarding pharmacokinetics characteristics of mitotane as previously cited, we can hypothesize that subpopulations are to be considered concerning Cl. If we suppose that the population has some heterogeneity and also consists of several homogeneous subpopulations, a simple extension of mixed effects models is a finite mixture of mixed effects models. When this is not explained by categorical covariates such as genotypes, sex, associated treatment, status, etc., we can use a latent covariate. Its purpose is to be able to use an underlying categorical covariate that we assume exists but is unknown. Then, this unknown covariate is treated like a random variable, and the probability of each modality is an integral part of a statistical model and should be assessed as well:(3)log(Cl)=log(Clpop)+βCl,lcat2[if {lcat=2plcat1=ℙ(lcat=1)plcat2=ℙ(lcat=2)=1−plcat1 ]+ηCl,

In this example, for clearance, there are two modalities (1 and 2). For each individual, only the probability for each modality is estimated. We used it to model the statistical mixture of the population on Cl and V parameters.
(4)$$\mathrm{BIC}\;=\;-2\log\;\mathrm{likelihood}\;\left(\mathrm{OFV}\right)\;+\;n_{p}\;\left(\mathrm{number}\;\mathrm{of}\;\mathrm{parameters}\right)\times \ln\left(N\right)\left[N\;\mathrm{for}\;\mathrm{observed}\;\mathrm{data}\right].$$

The covariate model was built using a stepwise procedure with forward inclusion and backward deletion. A covariate was kept in the model if it improved the fit, reduced interpatient variability, and decreased the Bayesian information criterion (BIC).

The addition of covariates was stopped when no more decrease of BIC was obtained. The statistical significance of covariate was individually evaluated during the stepwise deletion using the likelihood ratio test (LRT). The covariate was kept in the model if the LRT was significant (*P* < 0.05) when it was removed from the full model. In the final model, the 95% confidence interval of each parameter was estimated using bootstrapping (*n* = 1000) [29,30] performed by R package Rsmlx [31].

### 2.6. Internal Evaluation of the Model

Evaluation of the model was based on goodness-of-fit plots, i.e., observations versus individual and populations predictions, weighted individual residuals versus individual predictions and time (IWRES), plots of normalized prediction distribution error (NPDE) versus population predictions and time [32]. The prediction-corrected visual predictive checks (pcVPC) were performed [33]. This plot shows the time course of the 2.5th, 50th, and 97.5th percentiles of the simulated profiles and compared with observed data.

### 2.7. External Evaluation of the Model

We externally evaluated the consistency of the final model by the following different ways: (1) median plasma concentrations published by Kerkhofs et al. [34] were compared to our simulated median plasma concentrations obtained in 1000 virtual patients (with median value of the covariates) from doses corresponding to this publication; and (2) PTAs published by Kerkhofs et al. [34] and Mauclère-Denost et al. [35] were compared to ours, obtained in 1000 virtual patients (with median value of the covariates) from doses corresponding to these publications.

### 2.8. Monte Carlo Simulation Assessment for Different Dose Regimens

Monte Carlo simulations were performed by Simulx (mlxR: R package version 4.0.6; Inria, Paris, France) based on the final PK model to generate 1000 PK profiles of mitotane for each candidate regimens. Simulated trough concentrations were obtained for each condition. One standard dose regimen (3 g/day) and three high-dose regimens were investigated (6, 9, 12 g/day). Moreover, we used a loading dose of 15 g/day for one month followed by a maintenance dose of 5 g/day. As immediately using high dose regimen (> 6 g /day) increases the probability of adverse drug reactions, we also simulated a progressive loading dose regimen.

## 3. Results

### 3.1. Subject Characteristics

Thirty-eight patients (27 men and 11 women), corresponding to 503 plasma concentrations, were included for model development on the condition that the date of therapeutic initiation was known with at least four samples. Demographic characteristics of the study population are provided in Table 1.

### 3.2. Basic Model Building

A one-compartment model with first order absorption and linear elimination was identified as the best model to describe the pharmacokinetics of mitotane (See Appendix A). The bioavailability and absorption rate constant were fixed at 35% and 24 day^−1^, respectively. This model allowed us to screen the covariates on individual parameters. A two-compartment model was better only regarding the BIC, but the RSE for the central compartment (*V*) and omega *V* were greater than 50%. The inclusion of the latent variable improved model prediction, goodness-of-fit plots and BIC (∆BIC = 34), residual distribution (PWRES, IWRES, NPDE), and estimation accuracy of parameters (RSE).

### 3.3. Covariate Analysis

Significant triglyceride, HDL, and latent covariate (lcat2) effects were observed on Clearance (Cl). It was included in the final model and was associated with a reduction in the unexplained interindividual variability (IIV) of Cl from 54.4 to 29.3%. PK parameters of the final model are reported in Table 2. All were accurately estimated, as highlighted by the small RSEs from observed Fisher information matrix. The results of bootstrap medians and 95% confidence intervals were consistent. The bootstrap analysis confirmed the reliability and robustness of the parameter and random effect estimates. The final model with covariates is considered as representative.

### 3.4. Internal Evaluation of the Final Model

The precision of the PK parameter estimates was less than 30% for the structural parameters, random effects, residual error, and latent probabilities except for βHDL (45.9%) and plcat_2 (50.1%). The goodness-of-fit plots of the final models are presented in Figure 1, Figure 2, and Figure 3. Observed and predicted concentrations of mitotane were matched well by visualizing individual weighted residual (IWRES) versus time after dose and individual predictions (Figure 2). No major systematic bias was observed for NPDE (Figure 2). The pc-VPC plot presented in Figure 3 indicated a good predictive performance of the model. It can be observed that the median tendency and the dispersion of the observations appear to be satisfactorily predicted by the model. Overall, the 2.5th, 50th, and 97.5th percentiles of observed concentrations were within the predicted 95% confidence interval of these percentiles.

### 3.5. External Evaluation of the Final Model

The adequacy between trough concentrations described in the study of Kerkhofs et al. [34] and the simulated trough concentrations using the same mean cumulative dose regimens (272 g for low-dose group and 440 for high-dose group) was good. For the low-dose group, in the end of the experiment at week 12, median plasma level was 10.6 mg/L (range 2.3–18.1 mg/L) in the study and 8.2 mg/L in our simulated data. For the high-dose group, median plasma level at week 12 was 14.2 mg/L (range 2.1–29.7 mg/L) in the study and 13.3 mg/L in our simulated data (see Appendix A
Appendix A).

We also compared PTA between our simulated data and two studies—Kerkhofs et al. [34] and Mauclère-Denost et al. [35]. For the first one, with a mean cumulative dose of 272 g and 440 g for 12 weeks, the PTA was 33% (4/12) and 50% (10/20), respectively. PTAs estimated with our model were 14.6% (146/1000) and 46.5% (465/1000) and not different compared to those previously studied (Pearson’s chi-squared test, *P-*value > 0.05). For the second one, with a mean dosing regimen of 4.5 g/day, the PTA was 45% (10/22) after three months of therapy. With a simulation of the same dose regimen, PTA was 33.1% (331/1000) (Pearson’s chi-squared test, *P-*value > 0.05). This model can thus be used to perform simulations.

### 3.6. Simulations of Dosage Regimens

Figure 4 shows simulated concentration-time profiles between 0 and 90 days calculated for 3, 6, 9, and 12 g/day (Figure 4a–d). The PTA at three months was 10, 55, 76, and 85 %, respectively. Four additional concentration-time profiles between 0 and 90 days were also simulated: 6 g/day for patients with severe hypertriglyceridemia (7 g/L plasma level, with median HDL) (Figure 4e) or elevated clearance (latent covariate modality 2:plcat_2 = 1, with median HDL and Tg level) (Figure 4f), a loading dose of 15 g/day during one month followed by 5 g/day (Figure 4g), and a progressive loading dose as recommended from adapted summary of product characteristics (3 g at day 1, 4.5 g at day 2, 6 g at day 3, 7.5 g at day 4, 9 g at day 5, 10.5 g at day 6, 12 g at day 7, 13.5 g at day 8, 15 g/day between day 9-30, 5 g/day between days 31–90) (Figure 4h). The PTA at three months was 63.1, 3, 69, and 65%, respectively. For the last two dosing regimens, the PTA at one month was 57 and 51%, respectively. For these two dosing regimens (loading dose and progressive loading dose), the percentages of patients above 20 mg/L at three months were 42 an 39%, respectively, and were lower compared to high-dose regimens (9 and 12 g/day, 58 and 73%, respectively, except 6 g/day, which was 30.4%). 

## 4. Discussion

Using non-linear mixed effects modelling on one-stage population approach, we qualified a population pharmacokinetic model of mitotane which can predict different levels of plasma concentrations up to 1500 days of exposition. This long duration of follow-up is necessary to capture the late steady-state of mitotane. Using the routine data aggregated from requests of physicians following currently recommended practices, like TDM, we found that mitotane concentrations were best described by a one-compartment model. To our knowledge, this is the second population model describing interindividual PK variability of mitotane but the first to highlight two subpopulations and a Tg/HDL covariate effect on Cl. The inclusion of the latent covariate improved model prediction: goodness-of-fit plots and BIC (∆BIC = 34), residual distribution (PWRES, IWRES, NPDE) and estimation accuracy of parameters (Appendix A). Kerkhofs et al. [15] modelled mitotane concentrations by using an iterative two-stage Bayesian approach [36]. These modelling procedure does not distinguish inter- and intra-individual variability. Consequently, the variance is overestimated which requires many samples to estimate individual parameters. As mitotane has high lipophilic characteristics, they qualified a three-compartment model with an absorption constant (Ka) estimate much slower than elimination constant (*K_elm_*) corresponding to a flip-flop phenomenon such as sustained-release formulation vs. immediate-release formulation which was not explicated. The estimated Ka was 0.005 h^−1^ which seems to be physiologically incompatible with published data [12,13,37]. Nevertheless, we investigated a structure model with three-compartment by fixing Ka but the precision of parameter estimations was out of predefined criteria (RSE > 200%). Moreover, the terminal phase seems challenging to capture because the central compartment measured concentrations would be below our limit of quantification [12]. The first to use a pharmacokinetic modelling approach to capture the interindividual variability of estimated parameters were Arshad et al. [16]. The originality of their work was to implement time-varying clearance to characterize autoinduction on a one-compartment model.

The major difficulty for this model was choosing a value of Ka close to a physiological reality knowing that mitotane is a very lipophilic molecule (logP = 6.08). Based on animal studies, Hermansson et al. [12] studied the pharmacokinetics of mitotane in minipigs and doses were given as pure substance in corn oil, since earlier studies demonstrated that bioavailability was improved when given in oil emulsion compared to tablets [13,37]. The estimated Ka was 3 h^−1^ (=72 day^−1^). In the study of Watson et al. [37], the estimated Ka in dogs was 2.07 h^−1^ (=49.9 day^−1^) when given in oil emulsion. Note that an allometric application for this constant is not appropriate concerning Ka [38]. We can use it in human pharmacokinetics, but these values for Ka reflect a facilitated administration in oil emulsion. Therefore, in order to be close to human reality and have a global vision, we gleaned Ka values from PK studies modelling lipophilic molecule such as amlodipine [39] (logP = 2.22), olmesartan [40] (logP = 2.98), hydroxychloroquine [41] (logP = 3.87), chloroquine [42] (logP = 5.28), and telmisartan [43] (logP = 6.66). The estimated Ka was 0.64 h^−1^ (=15.36 day^−1^), 2.02 h^−1^ (=48.5 day^−1^), 1.15 h^−1^ (=27.6 day^−1^), 4.7 h^−1^ (=112 day^−1^), and 0.571 h^−1^ (=13.7 day^−1^), respectively. Moreover, when Ka cannot be reasonably estimated, we used the recommendation of Wade et al. [44]: to fix it at a reasonable value, usually 0.7–1 h^−1^ (=16.8–24 day^−1^), and then estimate the other model parameters. Arshad et al. fixed Ka at 49.9 day^−1^ with regard to Moolenar study. We modelled by using Ka from 15.36 to 112 day^−1^ (15.36, 24, 49.9, 72, 112). As Moolenar et al. [13] highlighted the fact that Ka is decreased with tablets compared to oil emulsion, we fixed Ka at 24 day^−1^ even if Ka from 15.36 to 49.9 day^−1^ had no consequence on estimated parameters.

Previous studies reported bioavailability values of 30–40% [9], so we chose not to estimate it. It should be noted that bioavailability for a lipophilic drug is generally variable depending on food intake, but we fixed it to 35%.

Analysis of the covariates shows that an increase in Tg and HDL-cholesterol levels leads to a decrease in the clearance. As previously reported [45], our results confirm the impact of lipid composition on mitotane binding to lipoproteins. Mitotane content of lipoproteins is positively correlated with both cholesterol (*r* = 0.77) and triglyceride content (*r* = 0.59). This is explained by the ability of lipoproteins to sequester mitotane. We also observed a slight negative correlation with LDL cholesterol and clearance, but it was statistically non-significant (*P-*value = 0.056). Huang et al. [40] found a negative effect of Tg on Cl for telmisartan which seems to have a similar absorption and bioavailability profile to mitotane. At the end of this analysis, we have identified a subpopulation of ultrafast metabolizers by using mixture model with a latent covariate. D’Avolio et al. [23] have shown that regarding *CYP2B6* G516T polymorphism, GG patients had lower plasma concentrations than mutated patients during the first six months. Beyond six months of treatment, plasma concentrations of mitotane of GG vs. GT/TT were not statistically significantly different. This confirms the hypothesis put forward by Arshad et al. that autoinduction via PXR ligands induces the activity of a broad range of processes in drug metabolism [21,46]. In the case of CYP2B6, PXR could induce it more slowly for patients with GT/TT genotype than GG genotype

Because of sparse data in our study, autoinduction highlighted by Arshad et al. [16] did not improve our final model with our population (see Appendix A). However, the pharmacokinetic analysis provided comparable range of parameter estimates, Arshad et al. vs. ours: V 6086 L and Cl*_pop_* 75 L/day *vs.* V, 8900 L and Cl 70 L/day. Arshad et al. [16] explained a part of variability of V with BMI. Unfortunately, no covariate related to body (BMI, IBW, LBW, TBW) did not explained a part of variability of V in our population. However, we found Tg and HDL covariates as part of variability for Cl.

We then performed Monte Carlo simulations to evaluate the standard recommended dosing regimen (3 g/day). We also tested three other high-dose regimens (6, 9, 12 g/day) in order to evaluate the probability of target attainment. Our simulations provided several important results. First, the recommended dosing regimens at the initiation of therapy (2–3 g/day) reduce the probability to reach the target at three months. Even with high-dose regimen, 6 g/day, the PTA is not optimal. Concerning 9–12 g/day, the probability to reach the therapeutic index is acceptable, 76–85%, but the risk to have probability to be above 20 mg/L is very high (58 and 73 % respectively). We simulated patients with hypertriglyceridemia or high-clearance for 6 g/day dosing regimens. Compared to normal population, having hypertriglyceridemia >7 g/L increases the PTA of 12%. For patients with high-clearance, the PTA drops drastically to 3%. We also simulated loading regimen, 15 g/day the first one month then 5 g/day. For all of this high dosing regimens, 6, 9, 12 g/day and loading dose regimen, we can improve the PTA but we highly increase the risk to be above the threshold of toxicity: 20 mg/L. So, high-dose regimens must be used with caution. For example, Kerkhofs et al. conducted a study where patients had “high-dose regimens” (6 g/day). After 12 weeks of treatment, the mean dosing regimen was 5.2 g/day cause of toxicity. In another example, Di Paoli et al. [47] documented in details a case-report which demonstrated that TDM for mitotane is essential. However, even with TDM, the patient recovered a good quality of life six months after multiple reduction of dosing regimen. The poor tolerability of mitotane always leads to reduce dose or suspending therapy because of high frequency of side effects. The main problem is the strong diffusion of mitotane in adipose tissues and organs reflected by a large V conducting to a slow elimination from the body. So, we decided to increase dosing regimen must be done in a very rational way to avoid observing a residual concentration >20 mg/L which will probably cause the occurrence of adverse effects.

Our study has several limitations. First, patients had sparse PK data with a limited data in the absorptive phase. Bioavailability and absorption phase are dependent on food intake [12,13,37] but this information was not available. Our data were only partial on therapeutic combinations and clinical response. However, if we want to improve our knowledge of target and type of target, do we still have to consider total mitotane concentrations or only the free fraction? We must compile PK/PD data in order to build joint models and focus on biomarkers of interest. Tolerability of mitotane was not evaluated also because of partial data. Clinical studies are necessary to assess the clinical utility and safety of mitotane doses higher than currently recommended.

Model-based simulations suggest that recommended dose regimen at initiation therapy is insufficient to reach therapeutic index in a proper delay. The main difficulty in optimizing mitotane is to be able to control a part of its high variability. Personalized care of patients with adrenocortical carcinoma begins to be rationalized by decisional algorithm [48]. Population pharmacokinetics should be systematically integrated to improve the medical decision. In a context of high interindividual variability and an inability to adjust individual concentrations because of an unfavourable pharmacokinetics, we can use this model to estimate individual parameters. The only rational manner to target attainment and improved visibility of physicians is to combine therapeutic drug monitoring and use this model to simulate dosing regimens to optimize exposure associated with the best response and lower adverse effects.

## 5. Clinical Application

In order to increase the probability of target attainment, empirical adaptation seems to be ambitious, even with TDM. The case report of Di Paolo et al. is a good example [47]. Thanks to TDM, patients did not have CNS or gastrointestinal toxicities, but this took long time (six months). Using TDM combined with a PK model is also helpful. To demonstrate this, we have selected a patient A which was non-responder and in subtherapeutic zone and a patient B which was responder but presenting poor tolerability signs. Our final model allowed us to estimate individual parameters for each patient (Figure 5). By using simulX (mlxR package) we can simulate and get the best dose regimen following desired target (Figure 6). As you can see on this figure, using TDM and PK model for rational adaptation is crucial if clinicians want to target therapeutic index more rapidly.

## Figures and Tables

**Figure 1 pharmaceutics-11-00566-f001:**
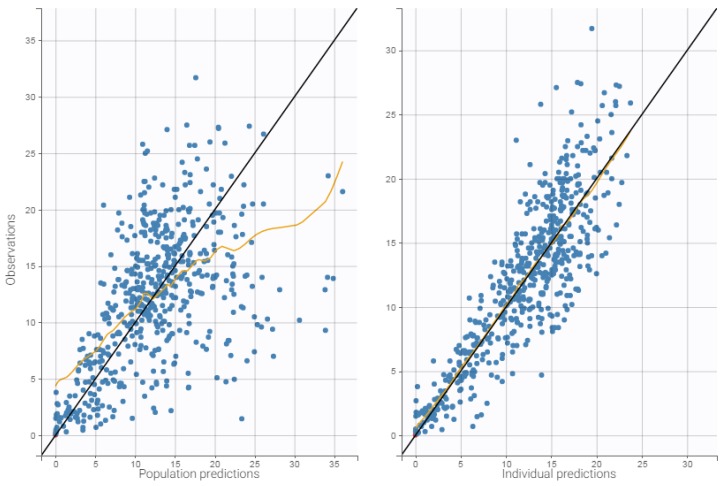
**Left**: Observed Mitotane versus population predictions; **Right**: Observed Mitotane versus individual predictions.

**Figure 2 pharmaceutics-11-00566-f002:**
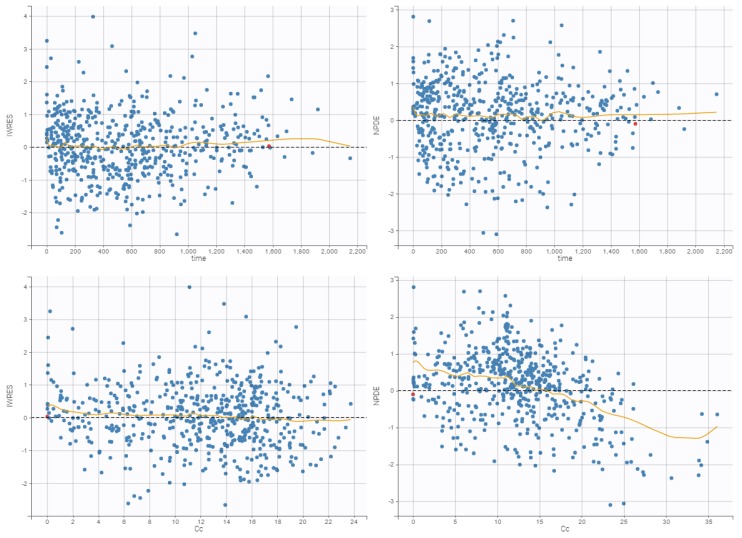
Individual weighted residual (IWRES) versus individual predictions (**top-left**) and time after dose (**bottom-left**). Normalized prediction errors (NPDE) versus population predictions (**top-right**) and time after dose (**bottom-right**). Abbreviations: Cc, concentrations of mitotane (mg/L); time in day.

**Figure 3 pharmaceutics-11-00566-f003:**
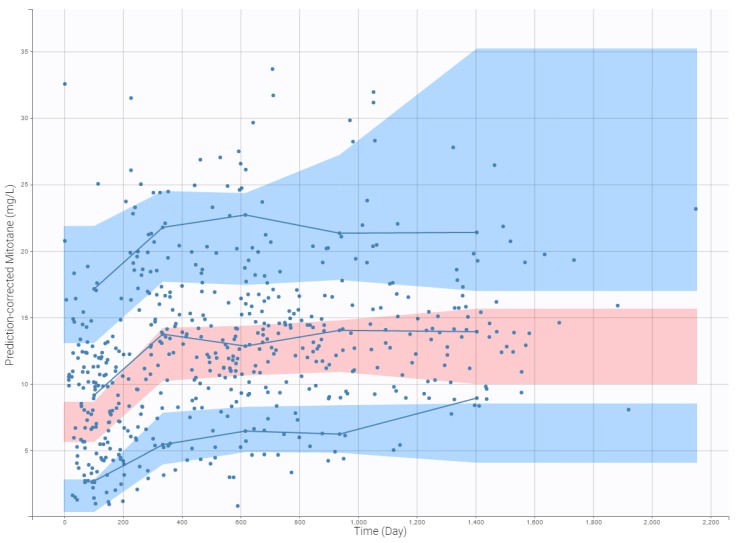
Prediction-corrected visual predictive check (pcVPC) for mitotane concentrations. Plot is utilized as part of model evaluation. Dots are observed mitotane concentrations, solid lines represent the median, 2.5th and 97.5th percentile of the observed values, and shaded areas represent the spread of 95% prediction intervals calculated from simulations.

**Figure 4 pharmaceutics-11-00566-f004:**
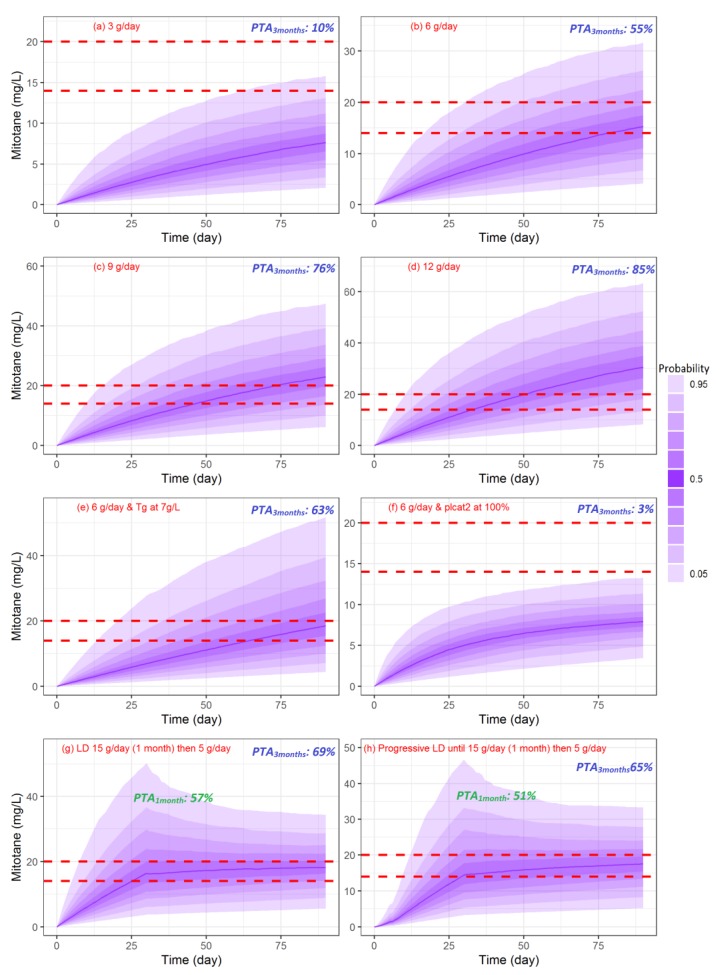
Simulated concentration profile of mitotane. (**a**), (**b**), (**c**), (**d**), (**g**), (**h**): with median covariates of our population except for (**e**) with Tg = 7 g/L and (**f**) with plcat2 = 100%.

**Figure 5 pharmaceutics-11-00566-f005:**
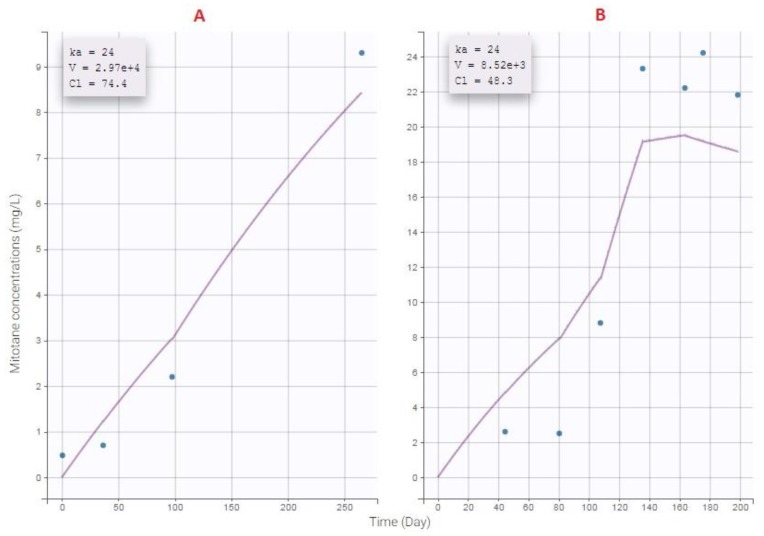
Estimated individual parameters of patients **A** and **B** with final model.

**Figure 6 pharmaceutics-11-00566-f006:**
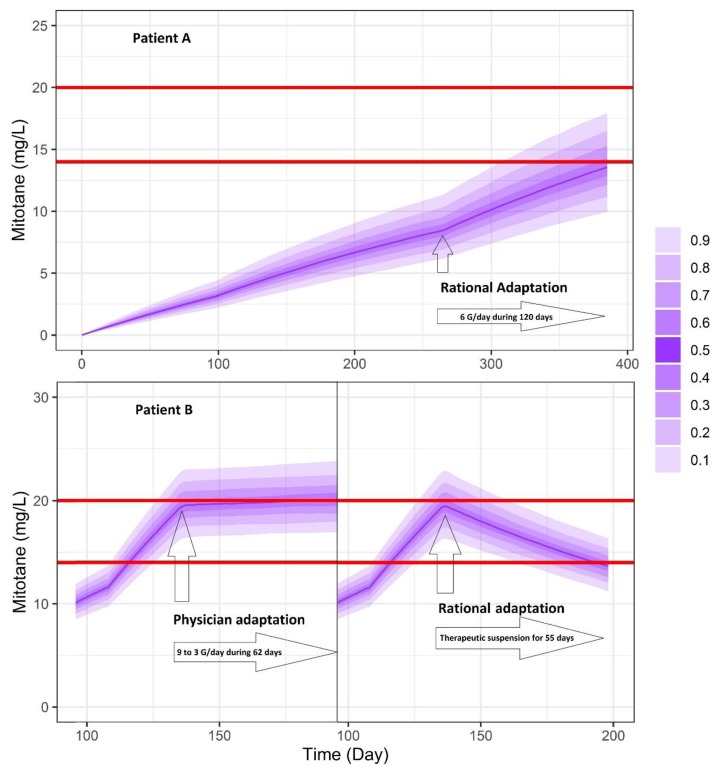
Rational adaptation dosing regimen of patients A and B with final model.

**Table 1 pharmaceutics-11-00566-t001:** Demographic characteristics and administration schedule of the 38 patients included in the PK analysis.

Parameter	Median (Mean, Standard Deviation; Min–Max)
Age (year)	51 (49.6, 14.5; 14–76)
Weight (kg)	71.7 (75.1, 19.1; 39–139)
BMI (kg/m²)	25.8 (27.3, 8; 15.8–56.4)
IBW (kg)	57.1 (60, 6.5; 49.4–71.9)
LBW (kg)	54.3 (53.9, 6.2; 35–63.8)
ClCR Gault-Cockcroft (mL/min)	103 (111.5, 43.3; 64.8–285.6)
HDL (g/L)	0.65 (0.73, 1.19; 0.26–2.1)
LDL (g/L)	1.61 (1.83, 1.99; 0.96–3.5)
TG (g/L)	1.56 (1.77, 1.14; 0.48–5.24)
Number of samples per patient	9 (13.2, 10; 4–46)
Amount of mitotane (g/day) per patient	2.9 (3.3, 1.3; 1–7.25) *

BMI, body mass index; IBW, ideal body weight; LBW, lean body weight; ClCR, Clearance of creatinine estimated Gault-Cockcroft formula; HDL, high density lipoprotein; LDL, low density lipoprotein; TG, Triglyceride. * Dosing regimen of eight patients were of 7.5–12 g/day during a part of their treatment.

**Table 2 pharmaceutics-11-00566-t002:** Estimates of the population pharmacokinetics parameters.

Parameter	Value (RSE %)	Median of Bootstrap ^†^ (95% CI)
**Fixed effects**		
Ka (day^−1^)	24 FIX	-
F (%)	35 FIX	-
V (L)	8900 (18.2)	9245 (6346–13467)
Cl (L·day^−1^)	70 (6.64)	68.4 (59.3–78.9)
βlcat2 *	1.12 (20.1)	1.12 (0.68–1.56)
βHDL **	−0.344 (45.9)	−0.334 (−0.644–−0.024)
βTg ***	−0.526 (25.5)	−0.516 (−0.779–−0.253)
**Between-subject variability**		
ω V(%)	90.4 (17.5)	89 (61–125)
ω Cl (%)	29.3 (16.7)	27.6 (19.6–39.1)
**Residual variability**		
a (constant)	1.06 (13.5)	1.05 (0.78–1.34)
b (proportional)	0.17 (8.51)	0.17 (0.14–0.20)
**Latent probabilities**		
plcat_1	0.885 (7.75)	0.875 (0.750–1.02)
plcat_2	0.115 (50.1)	0.125 (0.01–0.25)

Abbreviations are as follows: RSE, relative standard errors; CI, confidence interval. * Wald test: * *P*-value 6.5 × 10^−7^; ** *P*-value 0.029; *** *P*-value 8.8 × 10^−5^. ^†^ from 1000 bootstrap resampling. $Cl_{i}=Cl_{pop}\times {\left(\frac{Tg_{i}}{1.56}\right)}^{\beta_{Tg}^{Cl}}\times {\left(\frac{HDL_{i}}{0.65}\right)}^{\beta_{HDL}^{Cl}}\times e^{\beta_{Lcat2}^{Cl}}$.

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
