# Peer review of "Population Pharmacokinetics Modelling and Simulation of Mitotane in Patients with Adrenocortical Carcinoma: An Individualized Dose Regimen to Target All Patients at Three Months?"

_pharmaceutics, 2019, doi:10.3390/pharmaceutics11110566_

Round 1
Reviewer 1 Report
The authors built a PK model for mitotane via population modeling analysis. The manuscript could provide an informative insight into mitotane PK as a few studies on mitotane PK have been reported. However, the manuscript needs to be improved by considering the following comments:
Major comments:
1. Please describe the metabolism of mitotane before mentioning enzyme induction by mitotane.
2. The authors mentioned that mitotane could cause autoinduction of its metabolizing enzymes via PXR regulation. Has the autoinduction mechanism been considered in population PK modeling? What is the CL after single and multiple doses of mitotane? This mechanism should be considered if the CL after multiple doses is considerably higher than that after a single dose of mitotane.
3. Although the clinical data were collected retrospectively, the study should be approved by the corresponding clinical ethical review committee. Please clarify it.
4. It does not describe the mitotane administration route as well as the dosing-regimen of the study population. Please describe in more detail in the method section.
5. Please describe how CYP2B6 polymorphism was determined and show a number of patients per each genotype in a table.
6. In the method section, the authors said the confidential interval was 95%. However, the confidential interval in Figure 3 is 80%.
7. Page 9, what is the third dosing regimen? The first one is 6 g/day, and the second one is 15 g/day loading dose followed by 5 g/day. But, the third one is not clarified. Also, the authors said “For these dosing regimens, the percentage of patients above 20 mg/L was decreased compared to high-dose regimens (6, 9, 12 g/day)”. Please show the percentages of patients over 20 mg/L for each dosing regimen.
8. The authors did not show the results of models with time-varying CL. If mitotane causes autoinduction, the model with time-varying CL should be better then the model with linear elimination. Please show the results and discuss/compare them in the main text.
9. Page 12, lines 323-325; what is the criteria representing the adverse effect of mitotane in this modeling study? Is there any evidence from simulation that supports the authors’ claim? Please show evidence obtained from modeling and simulation.
10. Page 12, lines 3219-331; what were the specific simulation conditions for “hypertriglyceridemia” and “high clearance”?
11. Inter-individual variation for Vd is higher than 90%. Did the author perform covariate analysis for Vd? If increases in Tg and HDL levels lead to an increase of binding of mitotane with them, those covariates could show negative correlations with Vd.
Minor comments:
1. Page 2, line 45: What is ENS@T group? Is it correct?
2. Page 2, line 59: It should be “pregnane X receptor” not “pregnane R receptor”.
3. Please show PTA values for each dosing regimen on the figure for readers.
Author Response
October 25th, 2019
Dear Editor, Dear Reviewers
First, we want to thank you for your interest in our work and for giving us the opportunity to revise the manuscript. We also want to thank the referees for their helpful comments improving the quality of the manuscript.
We have addressed the Editor and Reviewers’ comments in the point-by-point response attached and improved the manuscript accordingly. We believe these changes significantly enhance the quality of the manuscript and hope that it now will be deemed suitable for publication in Pharmaceutics. Please find below our responses to you and the referees’ comments.
Best regards,
The authors
MDPI:
Authors:
Taking your remark into account, we have made changes wherever possible without changing the sense. For some classical formulations, we suggest not to amend the meaning too much.
Referee #1:
Major comments:
Please describe the metabolism of mitotane before mentioning enzyme induction by mitotane.
The authors: We added a paragraph on mitotane metabolism like you suggested:
Line 60. To identify metabolism-related variability, it has to be noted that mitotane is progressively metabolized in the liver by hydroxylation and oxidation to two compounds: o,p'-DDE and o,p'-DDA [18,19]. Kitamura et al. identified Cytochrome P450 isoforms which are implicated in the processus of dechlorination by microsomes from specific human cells expressing CYPs 1A1, 1A2, 2A6, 2B6, 2C9, 2D6, 2E1 and 3A4. It appeared that 3A4 and 2B6 were the most involved [19].
The authors mentioned that mitotane could cause autoinduction of its metabolizing enzymes via PXR regulation. Has the autoinduction mechanism been considered in population PK modeling? What is the CL after single and multiple doses of mitotane? This mechanism should be considered if the CL after multiple doses is considerably higher than that after a single dose of mitotane.
The autoinduction hypothesis is duly cited by Arshad et al., 2018 highlighting the modification of the CYP PXR regulation repressor. We tried using their model (Arshad et al.) or other equations (Equation S1) but, the results were not in favour of keeping time-varying clearance in our final model. In Table S1, we added information concerning Arshad et al.’s model and the best results with tested equations that is Equation 6 of table S1 to compare with ours.
Table S1. Summary of covariates model building and time-varying clearance.
Model |
Number of covariates |
-2LL |
BIC |
ΔBIC |
RSE of parameters |
Basic model (1 cmt) |
0 |
3005 |
3027 |
|
< 30% |
Tg on CL (1cmt) |
1 |
2994 |
3019 |
-8 |
< 30% |
Tg and HDL on CL (1cmt) |
2 |
2988 |
3015 |
-12 |
< 30% |
Tg, HDL and Lcat2 on CL (1cmt) |
3 |
2960 |
2093 |
-34 |
< 30% |
Basic model with TVC eq. 6 |
0 |
2999 |
3030 |
-3 |
> 1000% |
Arshad et al.[16] |
1 |
3026 |
3051 |
+24 |
< 30% |
Abbreviations are as follows: -2LL = -2 x loglikelihood; ΔBIC = BIC (model step) – BIC (basic model); RSE, Relative standard error; BIC, Bayesian information criterion; lcat2, latent covariate; Tg, triglyceride; TVC, Time-varying clearance.
Although the clinical data were collected retrospectively, the study should be approved by the corresponding clinical ethical review committee. Please clarify it.
The clinical ethical review committee was consulted and pronounced itself in favour of the analysis and the CNIL (“Commission Nationale de l’Informatique et des Libertés”) provided us with a registration number: “2049775 v 0”.
I clarified it in the manuscript: “This retrospective study was a population PK analysis of pooled data collected between 2008 and 2016 from patients treated with mitotane by oral route and total daily dose of 1-7.25 g/day in 2, 3 or 4 administrations. The two datasets were collected in patients treated for adrenocortical carcinoma in the university hospital of Reims (n = 25) and Montpellier (n = 13). Mitotane plasma concentrations were measured under recommended therapeutic drug monitoring conditions (TDM) as specified in summary of product characteristics. The inclusion criteria were: patients with ACC, with therapeutic initiation, with at least 4 blood samples and without missing data on covariates. The collection and use of computerized medical data were only possible after a declaration to the “Commission Nationale de l’Informatique et des Libertés” and obtaining a registration number: 2049775 v 0.”
It does not describe the mitotane administration route as well as the dosing-regimen of the study population. Please describe in more detail in the method section.
As suggested, we clarified the oral route and dosing regimen in the manuscript methods section. It is also specified in Table 1.
Please describe how CYP2B6 polymorphism was determined and show a number of patients per each genotype in a table.
CYP2B6 polymorphism was not determined in our study. We proposed it to partially explain the difference of clearance between groups of metabolizers. In the introduction of the manuscript, we describe the causes of important interindividual variability of mitotane, including this categorical variable since evoked by D’avolio et al. The distribution of Clearance parameter showed bimodal distribution. That is why we tested modeling mixture with latent covariate. We described this in the covariate analysis part. In the results, we have only 11.5% of patients with high clearance which are probably among the 74% of wild type patients (GG type). The inclusion of the latent covariate improved model prediction: goodness-of-fit plots and BIC (∆BIC=34), residual distribution (PWRES, IWRES, NPDE) and estimation accuracy of parameters (RSE).
In the method section, the authors said the confidential interval was 95%. However, the confidential interval in Figure 3 is 80%.
This may be confusing. We mentioned this confidence interval for each parameter of the final model determined from 1000 nonparameter bootstraps. The pcVPC plot shows 10th, 50th, 90th percentiles with shaded areas 90% prediction intervals calculated from simulations. To limit confusion, we modified the figure and the text as suggested:
The pc-VPC plot presented in Figure 3 indicated a good predictive performance of the model. It can be observed that the median tendency and the dispersion of the observations appear to be satisfactorily predicted by the model. Overall, the 2.5th, 50th and 97.5th percentiles of observed concentrations were within the predicted 95 % confidence interval of these percentiles.
Fig. 3 Prediction-corrected visual predictive check (pcVPC) for mitotane concentrations. Plot is utilized as part of model evaluation. Dots are observed mitotane concentrations, solid lines represent the median, 2.5th and 97.5th percentile of the observed values and shaded areas represent the spread of 95% prediction intervals calculated from simulations the median, 10th and 90th percentile.
Page 9, what is the third dosing regimen? The first one is 6 g/day, and the second one is 15 g/day loading dose followed by 5 g/day. But, the third one is not clarified. Also, the authors said “For these dosing regimens, the percentage of patients above 20 mg/L was decreased compared to high-dose regimens (6, 9, 12 g/day)”. Please show the percentages of patients over 20 mg/L for each dosing regimen.
We clarified this in the text for less confusion. Figure 4 shows simulated concentration-time profiles between 0 and 90 days calculated for 3, 6, 9 and 12 g/day (Fig 4. a, b, c, d). The PTA at 3 months was 10, 55, 76 and 85 %, respectively. Four additional concentration-time profiles between 0 and 90 days were simulated: 6 g/day for patients with severe hypertriglyceridemia (7 g/L) (Fig. 4e) or elevated clearance (latent covariate modality 2) (Fig. 4f), a loading dose of 15 g/day during one month followed by 5 g/day (Fig. 4g), and a progressive loading dose as recommended from adapted summary of product characteristics (3 g at day 1, 4.5 g at day 2, 6 g at day 3, 7.5 g at day 4, 9 g at day 5, 10.5 g at day 6, 12 g at day 7, 13.5 g at day 8, 15 g/day between day 9-30, 5 g/day between day 31-90) (Fig. 4h). The PTA at 3 months was 63.1, 3, 69, 65 %, respectively. For the last two-dosing regimens, the PTA at 1 month was 57 and 51, respectively.
#Please show the percentages of patients over 20 mg/L for each dosing regimen.
We did a mistake concerning the percentage of patients with dosing regimen 6g/day above 20 mg/L that is less than patients with loading dose regimens. We corrected and precised it :
“For these two dosing regimens (Loading dose and progressive loading dose), the percentages of patients above 20 mg/L at 3 months, were 42 an 39% respectively, and were lower compared to high-dose regimens: (6, 9 and 12 g/day, 58 and 73%, respectively except 6 g/day which was 30.4%.”
The authors did not show the results of models with time-varying CL. If mitotane causes autoinduction, the model with time-varying CL should be better then the model with linear elimination. Please show the results and discuss/compare them in the main text.
As previously said, autoinduction is highly plausible but in our case, the estimation of parameters is less satisfying regarding graph plots, RSE and BIC. Based on the modelling criteria, the basic model was better. See modified table S1 put in the modified manuscript:
Table S1. Summary of covariates model building and time-varying clearance.
Model |
Number of covariates |
-2LL |
BIC |
ΔBIC |
RSE of parameters |
Basic model (1 cmt) |
0 |
3005 |
3027 |
|
< 30% |
Tg on CL (1cmt) |
1 |
2994 |
3019 |
-8 |
< 30% |
Tg and HDL on CL (1cmt) |
2 |
2988 |
3015 |
-12 |
< 30% |
Tg, HDL and Lcat2 on CL (1cmt) |
3 |
2960 |
2093 |
-34 |
< 30% |
Basic model with TVC eq. 6 |
0 |
2999 |
3030 |
-3 |
> 1000% |
Arshad et al.[16] |
1 |
3026 |
3051 |
+24 |
< 30% |
Abbreviations are as follows: -2LL = -2 x loglikelihood; ΔBIC = BIC (model step) – BIC (basic model); RSE, Relative standard error; BIC, Bayesian information criterion; lcat2, latent covariate; Tg, triglyceride; TVC, Time-varying clearance.
The discussion was also updated:
“Because of sparse data in our study, autoinduction highlighted by Arshad et al. [16] did not improve our final model with our population (see Table S1). However, the pharmacokinetic analysis provided comparable range of parameter estimates, Arshad et al. vs. ours: V 6086 L and CLpop 75 L/day vs.V, 8900 L and CL 70 L/day. Arshad et al. [16] explained a part of variability of V with BMI. Unfortunately, no covariate related to body (BMI, IBW, LBW, TBW) did not explained a part of variability of V in our population. However, we found Tg and HDL covariates as part of variability for CL.”
Page 12, lines 323-325; what is the criteria representing the adverse effect of mitotane in this modeling study? Is there any evidence from simulation that supports the authors’ claim? Please show evidence obtained from modeling and simulation.
Based in reported studies, when concentrations are above 20 mg/L (1), the probability of adverse drug effect is high. That is why our criteria for adverse effects is the threshold of 20 mg/L, even if a few patients show good tolerability until 30 mg/L (2).
Simulation of different dosing regimens allows us to know if they are applicable in clinical setting. If the probability to of being above 20 mg/L is high, the dosing regimen will not be used in real situations. That is the purpose of simulations. The best thing to do to improve the prediction of occurrence of adverse effects is to combine the pharmacokinetic model? with a toxicity model. As of today, there is no model of toxic effects described yet and our data were too fragmentary on the collection of adverse effects over time to use it them.
For more precision we updated the sentences: “Concerning 9 – 12 g/day, the probability to reach the therapeutic index is acceptable, 76-85%, but the risk to have probability to be above 20 mg/L is very high (58 and 73 % respectively)”.
(1) Schteingart, D.E.; Doherty, G.M.; Gauger, P.G.; Giordano, T.J.; Hammer, G.D.; Korobkin, M.; Worden, F.P. Management of patients with adrenal cancer: recommendations of an international consensus conference. Endocr. Relat. Cancer 2005, 12, 667–680.
(2) Hermsen, I.G.; Fassnacht, M.; Terzolo, M.; Houterman, S.; den Hartigh, J.; Leboulleux, S.; Daffara, F.; Berruti, A.; Chadarevian, R.; Schlumberger, M.; et al. Plasma Concentrations of o,p′DDD, o,p′DDA, and o,p′DDE as Predictors of Tumor Response to Mitotane in Adrenocortical Carcinoma: Results of a Retrospective ENS@T Multicenter Study. J. Clin. Endocrinol. Metab. 2011, 96, 1844–1851.
Page 12, lines 329-331; what were the specific simulation conditions for “hypertriglyceridemia” and “high clearance”?
We decided to show the effect of covariates on mitotane concentration predictions by fixing TG with 7 g/L and Lcat2 with plcat2 100% (so CLi multiplied by e1.12 = high-clearance). The goal is to identify patients with these covariates to better manage the adaptation of the dosing regimen. The manuscript is updated accordingly: “Four additional concentration-time profiles between 0 and 90 days were also simulated: 6 g/day for patients with severe hypertriglyceridemia (7 g/L plasma level, with median HDL) (Fig. 4e) or elevated clearance (latent covariate modality 2 : plcat_2 =1, with median HDL and Tg level) (Fig. 4f),..”
Inter-individual variation for Vd is higher than 90%. Did the author perform covariate analysis for Vd? If increases in Tg and HDL levels lead to an increase of binding of mitotane with them, those covariates could show negative correlations with Vd.
In our population we did not find any significant effect of covariate on Vd, although Arshad et al. found a positive effect on Vd with BMI, and Kherkofs et al. a negative effect with LBM. We screened potential effects on Vd with IBW, TBW and LBM but without success. For negative effect on Vd with Tg and HDL, you are right, we can see it on Vd only if we parametrized the model with elimination constant, but this parametrization is less better than clearance parametrization.
Minor comments:
1. Page 2, line 45: What is ENS@T group? Is it correct?
2. Page 2, line 59: It should be “pregnane X receptor” not “pregnane R receptor”.
3. Please show PTA values for each dosing regimen on the figure for readers.
EnS@T group is the European Network for the Study of @drenal Tumors. More than 20 countries are a part of it. t gathers the best medical and scientific teams involved in adrenal tumors study. Done Done

Reviewer 2 Report
The manuscript of Cazaubon and colleagues reports the development and validation of a POP/PK model to investigate PK of mitotane and to propose dose individualisation in order to allow the attainment of target concentrations within the first 3 months of therapy. The topic is very interesting, because the paucity of data for this drug and for the particular PK characteristics of mitotane. Authors found that known covariates (together with a latent one) were capable to explain inter-individual variability in mitotane PK. Although the interest, some points need a discussion.
First of all, it is not completely clear how the clinical data were collected along the observation period. For example, the reader may guess that body weight was obtained at every time point when a plasma sample was withdrawn. Is it correct?
Second, the inter-occasion variability could solve some problems regarding the unexplained variability, but, as the Authors stated, to implement this analysis a richer database is required. For example, there is a recent case report showing that toxic effects of mitotane required a significant decrease in drug daily doses to be achieve a good tolerability together with a long-term control of the disease (Di Paolo et al, J Chemother 2019). And this point leads to the third point of discussion, because I have some concerns to propose higher mitotane doses to physicians, even if for short periods. Indeed, toxic effects could be relevant for patients in terms of both severity and duration, the latter due to the large volume of distribution of the drug. Indeed, once the drug has diffused across tissues and organs, then stopping mitotane administration can be associated to a slow decline in plasma concentrations. Therefore, I suggest to stress the fact that higher doses could be used with caution.
The last comment for the authors from my side is related to the therapeutic drug monitoring. I believe that the model presented in the manuscript is interesting a well as the simulated regimens, but I would read a brief comment about mitotane TDM as a current tool that helps caregivers to personalise standard and alternative regimens, as those proposed by the authors.
Minor
Please change “tolerance” with “tolerability” and, consequently, “intolerance” with “poor tolerability”.
Author Response
October 25th, 2019
Dear Editor, Dear reviewers
First, we want to thank you for your interest in our work and for giving us the opportunity to revise the manuscript. We also want to thank the referees for their helpful comments improving the quality of the manuscript.
We have addressed the Editor and Reviewers’ comments in the point-by-point response attached and improved the manuscript accordingly. We believe these changes significantly enhance the quality of the manuscript and hope that it now will be deemed suitable for publication in Pharmaceutics. Please find below our responses to you and the referees’ comments.
Best regards,
The authors
MDPI:
Authors:
Taking your remark into account, we have made changes wherever possible without changing the sense. For some classical formulations, we suggest not to amend the meaning too much.
Referee #2:
First of all, it is not completely clear how the clinical data were collected along the observation period. For example, the reader may guess that body weight was obtained at every time point when a plasma sample was withdrawn. Is it correct?
Yes it is. Data was collected throughout the treatment for each patient. Slight variation in covariates was observed. We had implemented covariates as a regressor to take into account this variability and to shunt use of inter-occasion variability but the result was not fulfilled diagnostic criteria. The median values of each covariate for each individual were taken into account in the modeling.
We mentioned it in the part “covariate analysis”: “As variation of covariates were not significate, we took the median of each for each individual.”
Second, the inter-occasion variability could solve some problems regarding the unexplained variability, but, as the Authors stated, to implement this analysis a richer database is required. For example, there is a recent case report showing that toxic effects of mitotane required a significant decrease in drug daily doses to be achieve a good tolerability together with a long-term control of the disease (Di Paolo et al, J Chemother 2019). And this point leads to the third point of discussion, because I have some concerns to propose higher mitotane doses to physicians, even if for short periods. Indeed, toxic effects could be relevant for patients in terms of both severity and duration, the latter due to the large volume of distribution of the drug. Indeed, once the drug has diffused across tissues and organs, then stopping mitotane administration can be associated to a slow decline in plasma concentrations. Therefore, I suggest to stress the fact that higher doses could be used with caution.
We are totally agree with you. Your case TDM example is a very good illustration of our remarks in clinical application. Of course there is certainly inter-occasion because of long-term period study. That’s why individual paramaters of patients has to be reevaluated each time. In your example like ours, patient B, beyond TDM, optimization with rational adaption allow to decrease plasma concentrations more rapidly. In Di Paoli et al.’s case, clinicians decided to reduce dosing regimen by 50% but it is not sufficient, patient continues suffering of CNS and gastrointestinal toxicities. Between toxicities and improving symptoms, it took 6 months. TDM is effective but the control of toxicities was very long. In our example toxicity patient B, with rational adaptation, we accelerated the processus of control. Because of very large Vd which is often associated with long elimination, rational control is necessary if we don’t want to lose time for patients.
We added in discussion:” We also simulated loading regimen, 15 g/day the first one month then 5 g/day. For all of this high dosing regimens, 6, 9, 12 g/day and loading dose regimen, we can improve the PTA but we highly increase the risk to be above the threshold of toxicity: 20 mg/L. So, high-dose regimens have to be used with caution. For example, Kerkhofs et al. conducted a study where patients had “high-dose regimens” (6 g/day). After 12 weeks of treatment, the mean dosing regimen was 5.2 g/day cause of toxicity. In another example, Di Paoli et al. [47] documented in details a case-report which demonstrated that TDM for mitotane is essential. But even with TDM, the patient recovered a good quality of life 6 months after multiple reduction of dosing regimen. The poor tolerability of mitotane always leads to reduce dose or suspending therapy because of high frequency of side effects. The main problem is the strong diffusion of mitotane in adipose tissues and organs reflected by a large Vd conducting to a slow elimination from the body. So, decided to increase dosing regimen must be done in a very rational way to avoid observing a residual concentration > 20 mg/L which will probably cause the occurrence of adverse effects.”
The last comment for the authors from my side is related to the therapeutic drug monitoring. I believe that the model presented in the manuscript is interesting a well as the simulated regimens, but I would read a brief comment about mitotane TDM as a current tool that helps caregivers to personalise standard and alternative regimens, as those proposed by the authors.
We added brief comment in clinical application : “In order to increase the probability of target attainment, empirical adaptation seems to be ambitious, even with TDM. The case report of Di Paolo et al. is a good example [47]. Thanks to TDM, patients did not have CNS or gastrointestinal toxicities but this took long time, 6 months. Using TDM combined with a PK model is also helpful. To demonstrate this, we have selected a patient A which was non-responder and in subtherapeutic zone and a patient B which was responder but presenting poor tolerability signs. Our final model allowed us to estimate individual parameters for each patient (Figure 5). By using simulX (mlxR package) we can simulate and get the best dose regimen following desired target (Figure 6). As you can see on this figure, using TDM and PK model for rational adaptation is crucial if clinicians want to target therapeutic index more rapidly.”
Minor
Please change “tolerance” with “tolerability” and, consequently, “intolerance” with “poor tolerability”.
Authors: Done.

Round 2
Reviewer 1 Report
The manuscript is improved and described the method and result more clearly.